# PREDICTING AUCTION PRICE OF VEHICLE LICENSE PLATE WITH DEEP RECURRENT NEURAL NETWORK

## ABSTRACT

In Chinese societies, superstition is of paramount importance, and vehicle license plates with desirable numbers can fetch very high prices in auctions. Unlike other valuable items, license plates are not allocated an estimated price before auction. I propose that the task of predicting plate prices can be viewed as a natural language processing (NLP) task, as the value depends on the meaning of each individual character on the plate and its semantics. I construct a deep recurrent neural network (RNN) to predict the prices of vehicle license plates in Hong Kong, based on the characters on a plate. I demonstrate the importance of having a deep network and of retraining. Evaluated on 13 years of historical auction prices, the deep RNN's predictions can explain over 80 percent of price variations, outperforming previous models by a significant margin. I also demonstrate how the model can be extended to become a search engine for plates and to provide estimates of the expected price distribution.

## 1 INTRODUCTION

Chinese societies place great importance on numerological superstition. Numbers such as 8 (representing prosperity) and 9 (longevity) are often used solely because of the desirable qualities they represent. For example, the Beijing Olympic opening ceremony occurred on 2008/8/8 at 8 p.m., the Bank of China (Hong Kong) opened on 1988/8/8, and the Hong Kong dollar is linked to the U.S. dollar at a rate of around 7.8.

License plates represent a very public display of numbers that people can own, and can therefore unsurprisingly fetch an enormous amount of money. Governments have not overlooked this, and plates of value are often auctioned off to generate public revenue. Unlike the auctioning of other valuable items, however, license plates generally do not come with a price estimate, which has been shown to be a significant factor affecting the sale price (Ashenfelter, 1989; Milgrom & Weber, 1982). The large number of character combinations and of plates per auction makes it difficult to provide reasonable estimates.

This study proposes that the task of predicting a license plate's price based on its characters can be viewed as a natural language processing (NLP) task. Whereas in the West numbers can be desirable (such as 7) or undesirable (such as 13) in their own right for various reasons, in Chinese societies numbers derive their superstitious value from the characters they rhyme with. As the Chinese language is logosyllabic and analytic, combinations of numbers can stand for sound-alike phrases. Combinations of numbers that rhyme with phrases that have positive connotations are thus desirable. For example, "168," which rhythms with "all the way to prosperity" in Chinese, is the URL of a major Chinese business portal (http://www.168.com). Looking at the historical data analyzed in this study, license plates with the number 168 fetched an average price of US$10,094 and as much as $113,462 in one instance. Combinations of numbers that rhyme with phrases possessing negative connotations are equally undesirable. Plates with the number 888 are generally highly sought after, selling for an average of $4,105 in the data, but adding a 5 (rhymes with "no") in front drastically lowers the average to $342.

As these examples demonstrate, the value of a certain combination of characters depends on both the meaning of each individual character and the broader semantics. The task at hand is thus closely

related to sentiment analysis and machine translation, both of which have advanced significantly in recent years.

Using a deep recurrent neural network (RNN), I demonstrate that a good estimate of a license plate's price can be obtained. The predictions from this study's deep RNN were significantly more accurate than previous attempts to model license plate prices, and are able to explain over 80 percent of price variations. There are two immediate applications of the findings in this paper: first, an accurate prediction model facilitates arbitrage, allowing one to detect underpriced plates that can potentially fetch for a higher price in the active second-hand market. Second, the feature vectors extracted from the last recurrent layer of the model can be used to construct a search engine for historical plate prices. Among other uses, the search engine can provide highly-informative justification for the predicted price of any given plate.

In a more general sense, this study demonstrates the value of deep networks and NLP in making accurate price predictions, which is of practical importance in many industries and has led to a huge volume of research. As detailed in the following review, studies to date have mostly relied on small, shallow networks. The use of text data is also rare, despite the large amount of business text data available. By demonstrating how a deep network can be trained to predict prices from sequential data, this study provides an approach that may improve prediction accuracy in many industrial applications.

## 2    LICENSE PLATE AUCTIONS IN HONG KONG

License plates have been sold through government auctions in Hong Kong since 1973, and restrictions are placed on the reselling of plates. Between 1997 and 2009, 3,812 plates were auctioned per year, on average.

Traditional plates, which were the only type available before September 2006, consist of either a two-letter prefix or no prefix, followed by up to four digits (e.g., AB 1, LZ 3360, or 168). Traditional plates can be divided into the mutually exclusive categories of special plates and ordinary plates. Special plates are defined by a set of legal rules and include the most desirable plates.[1] Ordinary plates are issued by the government when a new vehicle is registered. If the vehicle owner does not want the assigned plate, she can return the plate and bid for another in an auction. The owner can also reserve any unassigned plate for auction. Only ordinary plates can be resold.

In addition to traditional plates, personalized plates allow vehicle owners to propose the string of characters used. These plates must then be purchased from auctions. The data used in this study do not include this type of plate.

Auctions are open to the public and held on weekends twice a month by the Transport Department. The number of plates to be auctioned ranged from 90 per day in the early years to 280 per day in later years, and the list of plates available is announced to the public well in advance. The English oral ascending auction format is used, with payment settled on the spot, either by debit card or check.

## 3    RELATED STUDIES

Most relevant to the current study is the limited literature on the modeling price of license plates, which uses hedonic regressions with a larger number of handcrafted features (Woo & Kwok, 1994; Woo et al., 2008; Ng et al., 2010). These highly ad-hoc models rely on handcrafted features, so they adapt poorly to new data, particularly if they include combinations of characters not previously seen. In contrast, the deep RNN considered in this study learns the value of each combination of characters from its auction price, without the involvement of any handcrafted features.

The literature on using neural networks to make price predictions is very extensive and covers areas such as stock prices (Baba & Kozaki, 1992; Olson & Mossman, 2003; Guresen et al., 2011; de Oliveira et al., 2013), commodity prices (Kohzadi et al., 1996; Kristjanpoller & Minutolo, 2015; 2016), real estate prices (Do & Grudnitski, 1992; Evans et al., 1992; Worzola et al., 1995), electricity

---

[1]A detailed description of the rules is available on the government's official auction website: `http://www.td.gov.hk/en/public_services/auction_of_vehicle_registration_marks/how_to_obtain_your_favourite_vehicle_registration/schedule/index.html`.

prices (Weron, 2014; Dudek, 2016), movie revenues (Sharda & Delen, 2006; Yu et al., 2008; Zhang et al., 2009; Ghiassi et al., 2015), automobile prices (Iseri & Karlik, 2009) and food prices (Haofei et al., 2007). Most studies focus on numeric data and use small, shallow networks, typically using a single hidden layer of fewer than 20 neurons. The focus of this study is very different: predicting prices from combinations of alphanumeric characters. Due to the complexity of this task, the networks used are much larger (up to 1,024 hidden units per layer) and deeper (up to 9 layers).

The approach is closely related to sentiment analysis. A particularly relevant line of research is the use of Twitter feeds to predict stock price movements (Bollen et al., 2011; Bing et al., 2014; Pagolu et al., 2016), although the current study has significant differences. A single model is used in this study to generate predictions from character combinations, rather than treating sentiment analysis and price prediction as two distinct tasks, and the actual price level is predicted rather than just the direction of price movement. This end-to-end approach is feasible because the causal relationship between sentiment and price is much stronger for license plates than for stocks.

Finally, Akita et al. (2016) utilizes a Long-Short-Term Memory (LSTM) network to study the collective price movements of 10 Japanese stocks. The neural network in that study was solely used as a time-series model, taking in vectorized textual information from two simplier, non-neural-network-based models. In contrast, this study utilizies a neural network directly on textual information.

Deep RNNs have been shown to perform very well in tasks that involve sequential data, such as machine translation (Cho et al., 2014; Sutskever et al., 2014; Zaremba et al., 2014; Amodei et al., 2016) and classification based on text description (Ha et al., 2016), and are therefore used in this study. Predicting the price of a license plate is relatively simple: the model only needs to predict a single value based on a string of up to six characters. This simplicity makes training feasible on the relatively small volume of license plate auction data used in the study, compared with datasets more commonly used in training deep RNN.

## 4 MODELING LICENSE PLATE PRICE WITH A DEEP RECURRENT NEURAL NETWORK

The input from each sample is an array of characters (e.g., ["X," "Y," "1," "2," "8"]), padded to the same length with a special character. Each character $s_t$ is converted by a lookup table $g$ to a vector representation $\vec{h}_0^t$, known as *character embedding*:

$$g(s_t) = \vec{h}_0^t \equiv [h_{0,1}^t, ..., h_{0,n}^t]. \tag{1}$$

The dimension of the character embedding, $n$, is a hyperparameter. The values $h_{0,1}^t, ..., h_{0,n}^t$ are initialized with random values and learned through training. The embedding is fed into the neural network sequentially, denoted by the time step $t$.

The neural network consists of multiple bidirectional recurrent layers, followed by one or more fully connected layers (Schuster & Paliwal, 1997). The bidirectionality allows the network to access hidden states from both the previous and next time steps, improving its ability to understand each character in context. The network also uses batch normalization, which has been shown to speed up convergence (Laurent et al., 2016).

Each recurrent layer is implemented as follows:

$$\vec{h}_l^t = \left[ \vec{h}_{l-}^t : \vec{h}_{l+}^t \right], \tag{2}$$

$$\vec{h}_{l-}^t = f(B_l(W_{l-}\vec{h}_{l-1}^t + U_{l-}\vec{h}_{l-}^{t-1})), \tag{3}$$

$$\vec{h}_{l+}^t = f(B_l(W_{l+}\vec{h}_{l-1}^t + U_{l+}\vec{h}_{l+}^{t+1})), \tag{4}$$

$$B_l(\vec{x}) = \gamma_l \hat{x} + \vec{\beta}_l, \tag{5}$$

where $f$ is the rectified-linear unit, $\vec{h}_{l-1}^t$ is the vector of activations from the previous layer at the same time step $t$, $\vec{h}_l^{t-1}$ represents the activations from the current layer at the previous time step $t-1$, and $\vec{h}_l^{t+1}$ represents the activations from the current layer at the next time step $t+1$. $B$ is the

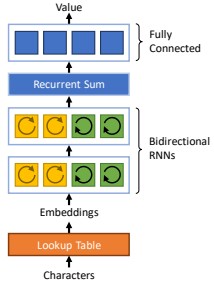

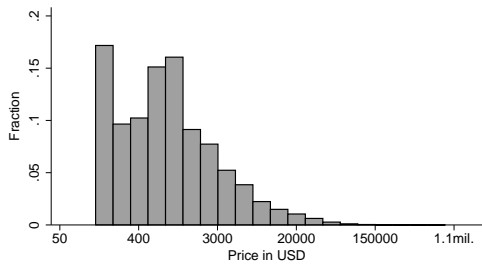

Figure 1: Sample Model Setup                Figure 2: Distribution of Plate Prices

BatchNorm transformation, and $\hat{x}$ is the within-mini-batch-standardized version of $x$.[2] $W$, $U$, $\gamma$ and $\beta$ are weights learnt by the network through training.

The fully connected layers are implemented as

$$\vec{h}_l = f(B_l(\vec{b}_l + W_l\vec{h}_{l-1})), \tag{6}$$

except for the last layer, which is implemented as

$$\vec{y} = \vec{b}_l + W_l\vec{h}_{l-1}. \tag{7}$$

$b_l$ is a bias vector learnt from training. The outputs from all time steps in the final recurrent layer are added together before being fed into the first fully connected layer.

To prevent overfitting, dropout is applied after every layer except the last (Hinton et al., 2012).

The model's hyperparameters include the dimension of character embeddings, number of recurrent layers, number of fully connected layers, number of hidden units in each layer, and dropout rate. These parameters must be selected ahead of training.

## 5 EXPERIMENT

### 5.1 DATA

The data used are the Hong Kong license plate auction results from January 1997 to July 2010, obtained from the HKSAR government. The data contain 52,926 auction entries, each consisting of i. the characters on the plate, ii. the sale price (or a specific symbol if the plate was unsold), and iii. the auction date.

Figure 2 plots the distribution of prices within the data. The figure shows that the prices are highly skewed: while the median sale price is $641, the mean sale price is $2,073. The most expensive plate in the data is "12," which was sold for $910,256 in February 2005. To compensate for this skewness, log prices were used in training and inference.

Ordinary plates start at a reserve price of HK$1,000 ($128.2), with $5,000 ($644.4) for special plates. The reserve prices mean that not every plate is sold, and 5.1 percent of the plates in the data were unsold. As these plates did not possess a price, we followed previous studies in dropping them from the dataset, leaving 50,698 entries available for the experiment.

The finalized data were divided into three parts, in two different ways: the first way divided the data randomly, while the second divided the data sequentially into non-overlapping parts. The second way creates a more realistic scenario, as it represents what a model in practical deployment would face. It is also a significantly more difficult scenario: because the government releases plates alphabetically through time, plates that start with later alphabets would not be available in sequentially-split data. For example, plates that start with "M" were not available before 2005, and plates that

---

[2] Specifically, $\hat{x}_i = \frac{x_i - \bar{x}_i}{\sqrt{\sigma_{x_i}^2 + \epsilon}}$, where $\bar{x}_i$ and $\sigma_{x_i}^2$ are the mean and variance of $x$ within each mini-batch. $\epsilon$ is a small positive constant that is added to improve numerical stability, set to 0.0001 for all layers.

start with "P" would not until 2010. It is therefore very difficult for a model trained on sequentially-split data to learn the values of plates starting with later alphabets.

In both cases, training was conducted with 64 percent of the data, validation was conducted with 16 percent, and the remaining 20 percent served as the test set.

## 5.2 TRAINING

I conducted a grid search to investigate the properties of different combinations of hyperparameters, varying the dimension of character embeddings $(12, 24, 48, 96, 128, 256)$, the number of recurrent layers $(1, 3, 5, 7, 9)$, the number of fully connected layers $(1, 3)$, the number of hidden units in each layer $(64, 128, 256, 512, 1024, 2048)$ and the dropout rate $(0, .05, .1)$. A total of 1080 sets of hyperparameters were investigated.

The grid search was conducted in three passes: In the first pass, a network was trained for 40 epochs under each set of hyperparameters, repeated 4 times. In the second pass, training was repeated 10 times for each of the 10 best sets of hyperparameters from the first pass, based on median validation RMSE. In the final pass, training was repeated for 30 times under the best set of hyperparameters from the second pass, again based on median validation RMSE. Training duration in the second and the third passes was 120 epochs.

During each training session, a network was trained under mean-squared error with different initializations. An Adam optimizer with a learning rate of 0.001 was used throughout (Kingma & Ba, 2014). After training was completed, the best state based on the validation error was reloaded for inference.

Training was conducted with four of NVIDIA GTX 1080s. To fully use the GPUs, a large mini-batch size of 2,048 was used.[3] During the first pass, the median training time on a single GPU ranged from 8 seconds for a 2-layer, 64-hidden-unit network with an embedding dimension of 12, to 1 minute 57 seconds for an 8-layer, 1,024-hidden-unit network with an embedding dimension of 24, and to 7 minutes 50 seconds for a 12-layer 2,048-hidden-unit network with an embedding dimension of 256.

Finally, I also trained recreations of models from previous studies as well as a series of fully-connected networks and character $n$-gram models for comparison. Given that the maximum length of a plate is six characters, for the $n$-gram models I focused on $n \leq 4$, and in each case calculated a predicted price based on the median and mean of $k$ closest neighbors from the training data, where $k = 1, 3, 5, 10, 20$.

## 5.3 MODEL PERFORMANCE

Table 1 reports the summary statistics for the set of parameters out of the 1080 sets specified in section 5.2, based on the median validation RMSE. The model was able to explain more than 80 percent of the variation in prices when the data was randomly split. As a comparison, *Woo et al. (2008)* and *Ng et al. (2010)*, which represent recreations of the regression models in (Woo et al., 2008) and (Ng et al., 2010), respectively, were capable of explaining only 70 percent of the variation at most.[4]

The importance of having recurrent layers can be seen from the inferior performance of the fully-connected network (MLP) with the same embedded dimension, number of layers and neurons as the best RNN model. This model was only capable of explaining less than 66 percent of the variation in prices.

In the interest of space, I include only two best-performing $n$-gram models based on median prices of neighbors. Both models were significantly inferior to RNN and hedonic regressions, being able to explain only 40 percent of the variation in prices. For unigram, the best validation performance was achieved when $k = 10$. For $n > 2$, models with unlimited features have very poor performance, as they generate a large number of features that rarely appear in the data. Restricting the number of

---

[3]I also experimented with smaller batch sizes of 64 and 512. By keeping the training time constant, the smaller batch size resulted in worse performance, due to the reduction in epochs.

[4] To make the comparison meaningful, the recreations contained only features based on the characters on a plate. Extra features such as date and price level are examined in Part ??.

Table 1: Model Performance

| Configuration | Train RMSE | Valid RMSE | Test RMSE | Train $R^2$ | Valid $R^2$ | Test $R^2$ |
|---|---|---|---|---|---|---|
| *Random Split* | | | | | | |
| RNN 512-128-5-2-.05 | .4391 | .5505 | .5561 | .8845 | .8223 | .8171 |
| Woo et al. (2008) | .7127 | .7109 | .7110 | .6984 | .7000 | .6983 |
| Ng et al. (2010) | .7284 | .7294 | .7277 | .6850 | .6842 | .6840 |
| MLP 512-128-7-.05 | .6240 | .6083 | .7467 | .78235 | .72785 | .6457 |
| unigram $k$NN-10 | .8945 | 1.004 | .9997 | .5221 | .4086 | .4088 |
| (1-4)-gram $k$NN-10 | .9034 | 1.012 | 1.013 | .5125 | .3996 | .3931 |
| | | | | | | |
| *Sequential Split* | | | | | | |
| RNN 512-48-5-2-.1 | .5018 | .5111 | .6928 | .8592 | .8089 | .6951 |
| Woo et al. (2008) | .7123 | .6438 | .8147 | .7163 | .6967 | .5783 |
| Ng et al. (2010) | .7339 | .6593 | .8128 | .6988 | .6819 | .5802 |
| MLP 512-48-7-.1 | .6326 | .6074 | .7475 | .7762 | .7300 | .6450 |
| unigram $k$NN-10 | .8543 | 1.046 | 1.094 | .5239 | .3979 | .3846 |
| (1-4)-gram $k$NN-10 | .8936 | 1.086 | 1.144 | .4791 | .3503 | .3269 |

Configuration of RNN is reported in the format of [Hidden Units]-[Embed. Dimension]-[Recurrent Layers]-[Fully Connected Layers]-[Dropout Rate]. Configuration of MLP is reported in the same format except there is no recurrent layer. Numbers for RNN, MLP and Ensemble models are the medians from 30 runs.

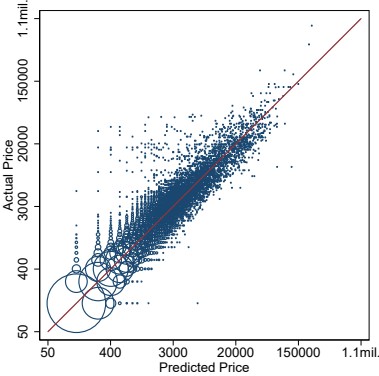

Figure 3: Actual vs Predicted Price
Plates are grouped by their predicted price and actual price, in bins of HK$1,000 ($128.2). The size of the circle represent the number of plates in a given bin.

Figure 4: Performance Fluctuations
The histogram represents the best model's test RMSE distribution. The red line is the kernel density estimate of the distribution. The two vertical lines indicate the validation RMSE of the comparison models.

features based on occurances and allowing a range of $n$ within a single model improve performance, but never surpassing the performance of the simple unigram. The performance of using median price and using mean price are very close, with a difference smaller than 0.05 in all cases.

All models took a significant performance hit when the data was split sequentially, with the RNN maintaining its performance lead over other models. The impact was particularly severe for the test set, because it was drawn from a time period furthest away from that of the train set.

Figure 3 plots the relationship between predicted price and actual price from a representative run of the best model, grouped in bins of HK$1,000 ($128.2). The model performed well for a wide range of prices, with bins tightly clustered along the 45-degree line. It consistently underestimated the price of the most expensive plates, however, suggesting that the buyers of these plates had placed on them exceptional value that the model could not capture.

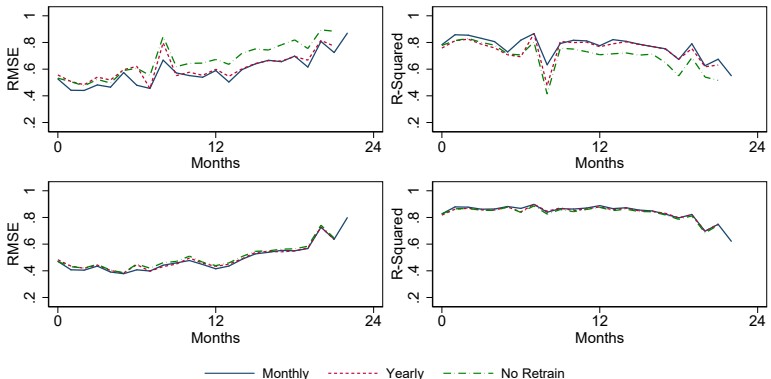

Figure 5: Impact of Retraining Frequency

## 5.4 MODEL STABILITY

Unlike hedonic regressions, which give the same predictions and achieve the same performance in every run, a neural network is susceptible to fluctuations due to convergence to local maxima. These fluctuations can be smoothed out by combining the predictions of multiple runs of the same model, although the number of runs necessary to achieve good results is then a practical concern.

Figure 4 plots the kernel density estimates of test RMSEs for the best models' 30 training runs. The errors are tightly clustered, with standard deviations of 0.025 for the randomly-split sample and 0.036 for the sequentially-split sample. This suggests that in practice several runs should suffice.

## 6 PERFORMANCE ENHANCEMENTS

### 6.1 RETRAINING OVER TIME

Over time, a model could conceivably become obsolete if, for example, taste or the economic environment changed. In this section, I investigate the effect of periodically retraining the model with the sequentially-split data. Specifically, retraining was conducted throughout the test data yearly, monthly, or never. The best RNN-only model was used, with the sample size kept constant at 25,990 in each retraining, which is roughly five years of data. The process was repeated 30 times as before.

Figure 5 plots the median RMSE and $R^2$, evaluated monthly. For the RNN model with no retraining prediction, accuracy dropped rapidly by both measures. RMSE increases an average of 0.017 per month, while $R^2$ dropped 0.01 per month. Yearly retraining was significantly better, with a 8.6 percent lower RMSE and a 6.9 percent higher $R^2$. The additional benefit of monthly retraining was, however, much smaller. Compared with the yearly retraining, there was only a 3.3 percent reduction in the RMSE and a 2.6 percent increase in the explanatory power. The differences were statistically significant.[5]

### 6.2 ENSEMBLE MODEL

Combining several models is known to improve prediction accuracy. This section considers a combination of the preceding neural network, (Woo et al., 2008) plus features not related to the characters on plates. The combination was conducted through linear regression, with the prediction of each

---

[5] Wilcoxon Sign-Rank Tests:
RNN yearly retraining = RNN no retraining: $z = -3.198, p = 0.001$
RNN monthly retraining = RNN yearly retraining: $z = -3.571, p = 0.000$
Combined yearly retraining = Combined no retraining: $z = -3.523, p = 0.000$
Combined monthly retraining = Combined yearly retraining: $z = -2.776, p = 0.006$

model acting as features. The model was thus implemented as follows:

$$y = \alpha + \delta_1 y_{rnn} + \delta_2 y_{woo} + \sum_i \nu_i x_i, \tag{8}$$

where $y_{rnn}$ is the prediction of the neural network, $y_{woo}$ the prediction of (Woo et al., 2008)'s regression model with only the license-plate-specific features, and $x_i$ a series of additional features, including the year and month of the auction, whether it was an afternoon session, the plate's position within the session's ordering, the existence of a prefix, the number of digits, a log of the local market stock index, and a log of the consumer price index. $\alpha$, $\delta$ and $\nu$ were estimated by linear regression on the training data.

For this ensemble model, the performance between different retraining frequencies was very close, with a less than 1 percent difference in the RMSE and a less than 2 percent difference in $R^2$ when going from no retraining to monthly retraining. Nevertheless, the differences remained statistically significant, as retraining every month did improve accuracy. The performance of the ensemble model was also considerably more stable than the RNN alone, with only half of the volatility at every retraining frequency. The primary reason behind this difference was the RNN's inability to account for extreme prices. The ensemble model was able to predict these extreme prices because (Woo et al., 2008) handcrafted features specifically for these valuable plates.

These results suggest that while there is a clear benefit in periodical retraining, this benefit diminishes rapidly beyond a certain threshold. Moreover, while deep RNN generally outperforms handcrafted features, the latter could be used to capture outliers.

## 7    EXPLAINING THE PREDICTIONS

Compared to models such as regression and $n$-gram it is relatively hard to understand the rationale behind a RNN model's prediction, given the large number of parameters involved and the complexity of the their interaction. If the RNN model is to be deployed in the field, it would need to be able to explain its prediction in order to convince human users to adopt it in practice. One way to do so is to extract a feature vector for each plate by summing up the output of the last recurrent layer over time. This feature vector is of the same size as the number of neurons in the last layer, which can be fed into a standard $k$-nearest-neighbor model to provide a "rationale" for the model's prediction.

To demonstrate this procedure, I use the best RNN model in Table 1 to generate feature vectors for all training samples. These samples are used to setup a $k$-NN model. When the user submit a query, a price prediction is made with the RNN model, while a number of examples are provided by the $k$-NN model as rationale.

Table 2 illustrate the outcome of this procedure with three examples. The model was asked to predict the price of three plates, ranging from low to high value. The predicted prices are listed in the *Prediction* section, while the *Historical Examples* section lists for each query the top four entries returned by the $k$-NN model. Notice how the procedure focused on the numeric part for the low-value plate and the alphabetical part for the middle-value plate, reflecting the value of having identical digits and identical alphabets respectively. The procedure was also able to inform the user that a plate has been sold before. Finally, the examples provided for the high-value plate show why it is hard to obtain an accurate prediction for such plates, as the historical prices for similar plates are also highly variable.

## 8    ESTIMATING THE DISTRIBUTION OF PRICES

While the RNN model outputs only a single price estimate, auctions that provide estimates typically give both a high and a low estimate. The $k$-NN model from the previous section can provide reasonably good estimates for common, low-value plates, but works poorly for rare, high-value plates due to the lack of similar plates in record. To tackle this problem, this section uses a Mixture Density Network (MDN) to estimate the distribution of plate prices (Bishop, 1994).

Table 2: Explaining Predictions with Automated Selection of Historical Examples

|  | Plate | Price | Plate | Price | Plate | Price |
|---|---|---|---|---|---|---|
| Prediction | LZ3360 | 1000 | MM293 | 5000 | 13 | 2182000 |
| Historical Examples | HC3360 | 1000 | MM293 | 5000 | 178 | 195000 |
|  | BG3360 | 3000 | MM203 | 5000 | 138 | 1100000 |
|  | HV3360 | 3000 | MM923 | 9000 | 12 | 7100000 |
|  | EC4360 | 1000 | MM296 | 4000 | 198 | 500000 |

The plates listed in the *Prediction* section are user queries and the prices are predictions. The plates and their corresponding prices listed in the *Historical Examples* section are historical data from the training sample.

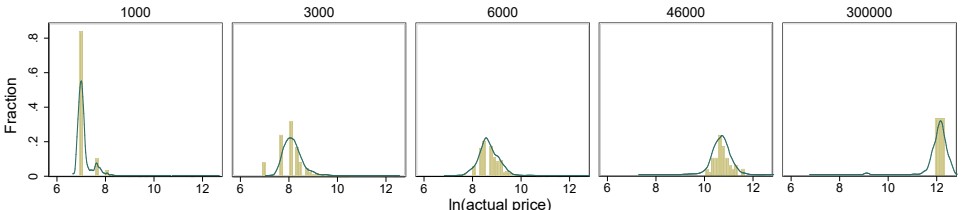

Figure 6: Estimated Density at Selected Predicted Prices

Bars and line represent the distribution of realized prices and the estimated density from the Mixture Density Network respectively.

The estimated probability distribution of realized price $p$ for a given predicted price $\hat{p}$ is

$$P(p \mid \hat{p}) = \sum_{k=1}^{24} \frac{e^{z_k(\hat{p})}}{\sum_{i=1}^{24} e^{z_i(\hat{p})}} \phi(p \mid \mu_k(\hat{p}), \sigma_k(\hat{p})), \tag{9}$$

where $[z_1(\hat{p}), ..., z_{24}(\hat{p}), \mu_1(\hat{p}), ..., \mu_{24}(\hat{p}), \sigma_1(\hat{p}), ..., \sigma_{24}(\hat{p})]$ is the output vector from a neural network with a single fully-connected layer of 256 neurons with a single input $\hat{p}$. The network was trained with the Adam optimizer for 5000 epochs, using the log likelihood of the distribution, $-\log\left(P(p \mid \hat{p})\right)$, as the cost function.

Figure 6 demonstrates the network's ability to fit the distribution of prices. The estimated density resembles the distribution of common, low-value plates, while producing a density that is noticeably wider than the distribution of actual prices for rare, high-value plates.

## 9 CONCLUDING REMARKS

This study demonstrates that a deep recurrent neural network can provide good estimates of license plate prices, with significantly higher accuracy than other models. The deep RNN is capable of learning the prices from the raw characters on the plates, while other models must rely on hand-crafted features. With modern hardware, it takes only a few minutes to train the best-performing model described previously, so it is feasible to implement a system in which the model is constantly retrained for accuracy.

A natural next step along this line of research is the construction of a model for personalized plates. Personalized plates contain owner-submitted sequences of characters and so may have vastly more complex meanings. Exactly how the model should be designed—for example, whether there should be separate models for different types of plates, or whether pre-training on another text corpus could help—remains to be studied.

## 10 ACKNOWLEDGMENTS

I would like to thank Travis Ng for providing the license plate data used in this study.

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
