# OpenReview forum: "Predicting Auction Price of Vehicle License Plate with Deep Recurrent Neural Network"
_ICLR.cc/2018/Conference — Reject_

### Official Review · AnonReviewer2 · 2017-11-27
**A successful application of RNN to a real value sentiment analysis task; well written but does not provide a novel network**

**Rating:** 6
**Confidence:** 5

**Review:**

The author(s) proposed to use a deep bidirectional recurrent neural network to estimate the auction price of license plates based on the sequence of letters and digits. The method uses a learnable character embedding to transform the data, but is an end-to-end approach. The analysis of squared error for the price regression shows a clear advantage of the method over previous models that used hand crafted features.
Here are my concerns:
1) As the price shows a high skewness in Fig. 1, it may make more sense to use relative difference instead of absolute difference of predicted and actual auction price in evaluating/training each model. That is, making an error of $100 for a plate that is priced $1000 has a huge difference in meaning to that for a plate priced as $10,000.

2) The time-series data seems to have a temporal trend which makes retraining beneficial as suggested by authors in section 7.2. If so, the evaluation setting of dividing data into three *random* sets of training, validation, and test, in 5.3 doesn't seem to be the right and most appropriate choice. It should however, be divided into sets corresponding to non-overlapping time intervals to avoid the model use of temporal information in making the prediction.

---

> ### Author Response · Authors · 2018-01-05
> **Response to Anonymous Review 2**
>
> Thank you for your comments and suggestions.
>
> 1. The use of log prices addresses this issue specifically. A $100 error for a $1000 plate increases the cost function exactly as much as a $100K error for a $1M plate.
>
> 2. The revised paper now includes the statistics from a sequential split of the data. Specifically, the oldest 64% of the data was used to train the model, the middle 16% was used for validation and the most recent 20% was used for testing. As explained in the paper, new plates were issued alphabetically by the government over time. Due to the lack of comparable plates in the training data, predicting the price of these new plates is very difficult for any model. Even so, the RNN model still maintain its performance lead over the other models.
>
> The performance penalty largely disappears if the most recent data is used for training and the oldest for testing. This is because the government routinely auction off plates that had been returned.

---

### Official Review · AnonReviewer3 · 2017-11-27
**RNNs applied to predicting auction prices from license plate numbers in China, Overall approach is not very scientific.**

**Rating:** 4
**Confidence:** 4

**Review:**

Summary: The authors take two pages to describe the data they eventually analyze - Chinese license plates (sections 1,2), with the aim of predicting auction price based on the "luckiness" of the license plate number.  The authors mentions other papers that use NN's to predict prices, contrasting them with the proposed model by saying they are usually shallow not deep, and only focus on numerical data not strings. Then the paper goes on to present the model which is just a vanilla RNN, with standard practices like batch normalization and dropout.  The proposed pipeline converts each character to an embedding with the only sentence of description being "Each character is converted by a lookup table to a vector representation, known as character embedding."   Specifics of the data,  RNN training, and the results as well as the stability of the network to hyperparameters is also examined. Finally they find a "a feature vector for each plate by summing up the output of the last recurrent layer overtime." and the use knn on these features to find other plates that are grouped together to try to explain how the RNN predicts the prices of the plates. In section 7,  the RNN is combined with a handcrafted feature model he criticized in a earlier section for being too simple to create an ensemble model that predicts the prices marginally better.

Specific Comments on Sections:
Comments: Sec 1,2
In these sections the author has somewhat odd references to specific economists that seem a little off topic, and spends a little too much time in my opinion setting up this specific data.

Sec 3
The author does not mention the following reference: "Deep learning for stock prediction using numerical and textual information" by Akita et al. that does incorporate non-numerical info to predict stock prices with deep networks.

Sec 4
What are the characters embedded with? This is important to specify. Is it Word2vec or something else? What does the lookup table consist of? References should be added to the relevant methods.

Sec 5
I feel like there are many regression models that could have been tried here with word2vec embeddings that would have been an interesting comparison. LSTMs as well could have been a point of comparison.

Sec 6
 Nothing too insightful is said about the RNN Model.

Sec 7
The ensembling was a strange extension especially with the Woo model given that the other MLP architecture gave way better results in their table.

Overall: This is a unique NLP problem, and it seems to make a lot of sense to apply an RNN here, considering that word2vec is an RNN. However comparisons are lacking and the paper is not presented very scientifically.  The lack of comparisons made it feel like the author cherry picked the RNN to outperform other approaches that obviously would not do well.

---

> ### Author Response · Authors · 2018-01-11
> **Response to Anonymous Review 3**
>
> Thank you for your detailed comments and suggestions. The following are improvements I have made:
> - The odd reference in the introduction was in response to a referee's inquiry in a previous submission. It has been removed. The introduction has also been shortened.
> - Citation has been added for Akita et al.
> - The construction of the character embeddings has been clarified. The lookup table maps each character into a different vector. The dimension of the vector is a hyperparameter. The elements of each vector are initialized with random values and learnt through training.
> - I have tried both both LSTM and GRU and find that they provided minimal improvement while lengthening the training time. The most likely reason is that plates are limited to at most 6 characters, which limit the impact of long-term dependency. Detailed comparison will be added in a future version of the paper.
> - Improvements to cross validation procedure and a new section on price distribution have been added.

---

### Official Review · AnonReviewer1 · 2017-11-28
**A nicely written paper on an interesting applied problem, but lacking some of the novelty and originality to be expected at a top conference.**

**Rating:** 4
**Confidence:** 4

**Review:**

The authors present a deep neural network that evaluates plate numbers. The relevance of this problem is that there are auctions for plate numbers in Hong Kong, and predicting their value is a sensible activity in that context. I find that the description of the applied problem is quite interesting; in fact overall the paper is well written and very easy to follow. There are some typos and grammatical problems (indicated below), but nothing really serious.

So, the paper is relevant and well presented. However, I find that the proposed solution is an application of existing techniques, so it lacks on novelty and originality. Even though the significance of the work is apparent given the good results of the proposed neural network, I believe that such material is more appropriate to a focused applied meeting. However, even for that sort of setting I think the paper requires some additional work, as some final parts of the paper have not been tested yet (the interesting part of explanations). Hence I don't think the submission is ready for publication at this moment.

Concerning the text, some questions/suggestions:
- Abstract, line 1: I suppose "In the Chinese society..."--- are there many Chinese societies?
- The references are not properly formatted; they should appear at (XXX YYY) but appear as XXX (YYY) in many cases, mixed with the main text.
- Footnote 1, line 2: "an exchange".
- Page 2, line 12: "prices. Among".
- Please add commas/periods at the end of equations.
- There are problems with capitalization in the references.

---

> ### Author Response · Authors · 2018-01-05
> **Response to Anonymous Review 1**
>
> Thank you for your thoughtful comments and suggestions.
>
> Taking your comments into consideration, the major improvement the revised paper has made is the use of a Mixture Density Model to estimate the distribution of realized price for a given predicted price. This is particularly useful for pricing rare, high-value plates, which has little historical data to speak of due to the lack of similar plates in records.
>
> I have also corrected all textual and formatting mistakes. The only exception is "Chinese societies"---in my view, there are indeed multiple Chinese societies. Take for example China, Taiwan and Hong Kong. Even though they share the same written language,  they each have their own culture and dialects that are distinct from the others.

---

### Decision · Program_Chairs · 2018-01-29
**ICLR 2018 Conference Acceptance Decision**

**Decision:**

Reject

**Comment:**

Reviewers concur that the paper and the application area are interesting but that the approaches are not sufficiently novel to justify presentation at ICLR.